# Novel Phosphorylated Penta-1,4-dien-3-one Derivatives: Design, Synthesis, and Biological Activity

**DOI:** 10.3390/molecules24050925

**Published:** 2019-03-07

**Authors:** Lijuan Chen, Tao Guo, Rongjiao Xia, Xu Tang, Ying Chen, Cheng Zhang, Wei Xue

**Affiliations:** State Key Laboratory Breeding Base of Green Pesticide and Agricultural Bioengineering, Key Laboratory of Green Pesticide and Agriculture Bioengineering, Ministry of Education, Guizhou University, Huaxi District, Guiyang 550025, China; 13765645934@163.com (L.C.); guotao9405@163.com (T.G.); xrjiao@163.com (R.X.); tx1173182020@163.com (X.T.); 15286114381@163.com (Y.C.); zhangcheng6954@163.com (C.Z.)

**Keywords:** phosphorylation, penta-1,4-dien-3-ones, antibacterial activities, antiviral activities, microscale thermophoresis, molecular docking

## Abstract

A series of novel phosphorylated penta-1,4-dien-3-one derivatives were designed and synthesized. The structures of all title compounds were determined by ^1^H-NMR, ^13^C-NMR, ^31^P-NMR, and high-resolution mass spectrometry (HRMS). Bioassay results showed that several of the title compounds exhibited remarkable antibacterial and antiviral activities. Among these, compound **3g** exhibited substantial antibacterial activity against *Xanthomonas oryzae pv. Oryzae* (Xoo), with a 50% effective concentration (EC_50_) value of 8.6 μg/mL, which was significantly superior to bismerthiazol (BT) (58.8 µg/mL) and thiodiazole-copper (TC) (78.7 μg/mL). In addition, compound **3h** showed remarkable protective activity against tobacco mosaic virus (TMV), with an EC_50_ value of 104.2 μg/mL, which was superior to that of ningnanmycin (386.2 μg/mL). Furthermore, the microscale thermophoresis and molecular docking experiments on the interaction of compounds **3h** and **3j** with TMV coat protein (TMV CP) were also investigated. Compounds **3h** and **3j** bound to TMV CP with dissociation constants of 0.028 and 0.23 μmol/L, which were better than that of ningnanmycin (0.52 μmol/L). These results suggest that novel phosphorylated penta-1,4-dien-3-one derivatives may be considered as an activator for antibacterial and antiviral agents.

## 1. Introduction

Bacterial diseases, such as rice bacterial leaf blight and citrus canker, caused by the pathogens *Xanthomonas oryzae pv. oryzae* (Xoo) and *Xanthomonas axonopodis pv. citri* (Xac) [1,2,3,4], strongly restrain the agricultural output worldwide and are difficult to control in agriculture. Furthermore, one of the most severe pathogenic viruses, the tobacco mosaic virus (TMV), can infect various crops, such as tobacco, pepper, cucumber, and other economic crops, which causes considerable crop loss [5,6,7]. To date, the few commercially available bactericides and plant virucides, such as thiodiazole-copper (TC), bismerthiazol (BT), ningnanmycin, and ribavirin, not only enhance resistance in the target pathogens, but are also detrimental for both the environment and plant health [8]. Therefore, developing new antibacterial and antiviral agents remains an important task for the medical community.

Pesticides based on natural products show more advantages than synthesized chemicals, e.g., low toxicity, simple decomposition, unique modes of action, and environmental friendliness [9,10,11,12]. Therefore, using bioactive natural product as leading molecules to design and synthesize pesticides is a developing trend. As an important analog of curcumin isolated from turmeric, penta-1,4-dien-3-one (Figure 1a) possesses numerous potential biological activities, including antibacterial [13], antiviral [14], antifungal [15], and antitumor [16] activities. In a previous study from our group, we synthesized a series of penta-1,4-dien-3-one derivatives, most of which exhibited excellent antiviral activities [13,17,18].

Phosphonates and their immediate derivatives possess a wide range of biological activities [19,20,21,22,23,24,25] and are, therefore, widely employed as plant virucides (Figure 1b), bactericides, fungicides, herbicides (Figure 1c), and plant growth regulators. Because the phosphorus–carbon bond of phosphonate derivatives is not susceptible to enzymatic degradation, they possess improved cell permeability, a lipophilic nature, and good physiological stability [26,27]. In a previous study from our group, we synthesized a series of phosphorylated flavonoid derivatives (Figure 2), most of which exhibited excellent antibacterial activities [21,28]. However, relatively few reports were conducted on plant antiviral activities. Therefore, phosphonate attracted considerate attention in the field of pesticide application.

Motivated by the abovementioned findings and to continue our efforts for developing highly efficient agrochemicals, we introduced a penta-1,4-diene-3-one group into phosphate derivatives, which might generate novel phosphate derivatives with potent biological activities. Thus, 19 novel phosphorylated penta-1,4-dien-3-one derivatives were designed, synthesized, and evaluated for their antibacterial activities against Xac and Xoo in vitro and their antiviral activity against TMV in vivo.

## 2. Results and Discussion

### 2.1. Chemistry

The synthetic route to novel phosphorylated penta-1,4-dien-3-one derivatives is depicted in Scheme 1. According to previous reports [15,29], 2-hydroxybenzaldehyde and acetone were used as the starting materials, and reacted with 10% NaOH for 12 h at ambient temperature to yield the intermediate (*E*)-4-(2-hydroxyphenyl)but-3-en-2-one (**1**). Intermediate **2** was obtained via condensation of intermediate **1** with substituted aldehydes. Finally, the title compounds **3a**–**3r** were synthesized by condensing intermediates **2** and dialkyl phosphonate in the presence of Et_3_N in CCl_4_ at ambient temperature for 24 h.

The structures of all title compounds were determined by ^1^H-NMR, ^13^C-NMR, ^31^P-NMR, and HRMS, and the spectra data are shown in the Appendix A. The representative data for **3a** are shown below. In ^1^H-NMR spectra, multiplet signals at δ 8.00–7.00 ppm indicate the presence of protons in olefinic bonds and aromatic nuclei, and two singlets at δ 3.87 and 3.85 ppm indicate the presence of –CH_3_ groups. Absorption signals at δ 188 and 55 ppm in ^13^C-NMR spectra confirm the presences of –C=O– and –CH_3_ groups, respectively. The high-resolution mass spectrometry (HRMS) spectra of target compounds show characteristic absorption signals of [M + H]^+^ ions, which is consistent with their molecular weight.

### 2.2. Antibacterial Activity of Title Compounds against Xoo and Xac In Vitro

The antibacterial activities of the target compounds **3a**–**3s** against two phytopathogenic bacterial (Xoo and Xac) were tested in vitro by the turbidimeter test [30,31,32]. Commercial agricultural antibacterial TC and BT were used as references, as shown in Table 1. Most of the compounds exhibited significant antibacterial activities against Xoo and Xac at 100 or 50 μg/mL.

Several of the compounds showed excellent activities against Xoo compared to TC and BT. Among these, the antibacterial activities of compounds **3a**, **3e**, **3g**, **3h**, and **3j** against Xoo at 100 μg/mL were 85.0, 69.4, 100.0, 86.5, and 85.5%, respectively, exceeding those of both TC (50.2%) and BT (64.9%). Activities of compounds **3a**, **3e**, **3g**, **3h**, and **3j** against Xoo at 50 μg/mL were 83.9, 65.3, 95.5, 78.1, and 81.4%, respectively, which were better than those of both TC (37.2%) and BT (45.2%). The antibacterial activities of compound **3e** against Xac at 100 and 50 μg/mL were 77.6 and 67.9%, respectively, which was superior to those of TC (57.2 and 27.8%) and BT (70.3 and 54.9%).

To further confirm the antibacterial activities of our target compounds, the EC_50_ values were tested for several compounds, and the results are listed in Table 2. Compounds **3a**, **3e**, **3g**, **3h**, and **3j** exhibited remarkable antibacterial activities against Xoo, with EC_50_ values of 35.8, 46.0, 8.6, 19.1, and 17.7 μg/mL, which were much better than those of BT (58.8 µg/mL) and TC (78.7 μg/mL). Compounds **3b** and **3e** exhibited excellent antibacterial activities against Xac, with EC_50_ values of 34.1 and 22.9 μg/mL, which were significantly superior to those of BT (44.5 μg/mL) and TC (87.9 μg/mL). In particular, compound **3e** exhibited excellent activities against both Xoo (46.0μg/mL) and Xac (22.9 μg/mL). These results indicate that those compounds should be further studied as potential alternative templates in the search for novel antibacterial agents. 

### 2.3. Structure–Activity Relationship (SAR) of Antibacterial Activities

As indicated in Table 1 and Table 2, most phosphorylated penta-1,4-dien-3-one derivatives showed significant antibacterial activities against Xoo, and some structure–activity relationships can be analyzed and summarized. Firstly, the presence of 4-Cl-Ph, 2-Cl-Ph, 3-NO_2_-Ph, and 4-NO_2_-Ph groups at the R_1_ position greatly improved the antibacterial activities of the title compounds against Xoo. For example, the title compounds **3a** (4-Cl-Ph), **3e** (2-Cl-Ph), **3g** (3-NO_2_-Ph), **3h** (3-NO_2_-Ph), and **3j** (4-NO_2_-Ph) exhibited significant antibacterial activities against Xoo at 100 μg/mL, with inhibition rates of 85.0, 69.4, 100.0, 86.5, and 85.5%, respectively. These compounds were found to be more active compared to other tested compounds. Secondly, with the presence of the –Cl-Ph group at the R_1_ position, the corresponding compounds presented better in vitro bioactivity against Xoo, following the order of **3a** (R_1_=4-Cl-Ph, R_2_ = CH_3_) > **3n** (R_1_=4-Br-Ph, R_2_=CH_3_), **3b** (R_1_=4-Cl-Ph, R_2_=CH_2_CH_3_) > **3o** (R_1_=4-Br-Ph, R_2_=CH_2_CH_3_). Thirdly, introducing a 3-NO_2_-Ph at the R_1_ position is favorable for the antibacterial activity of title compounds against Xoo. For instance, those compounds bearing a 3-NO_2_-Ph group (**3g** and **3h**) have better antibacterial activities against Xoo than those compounds containing a 4-NO_2_-Ph group (**3i** and **3j**). 

### 2.4. Antiviral Activity of Title Compounds against TMV In Vivo

Using *Nicotiana tabacum* L. leaves of the same age as the test subjects, the curative and protective activities against TMV in vivo at 500 μg/mL were evaluated by the half-leaf blight spot method [33,34]. The commercial agricultural antiviral agent ningnanmycin was used as a control, and the preliminary bioassays results are listed in Table 3. All tested compounds exhibited from weak to good antiviral activities against TMV. Among these, compounds **3a**, **3c**, **3l**, and **3s** exhibited excellent curative activities against TMV at 61.2, 62.2, 69.3, and 63.4%, respectively, which exceeded that of ningnanmycin (56.1%). The protective activities of **3h, 3j**, and **3p** (66.7, 63.8, and 58.8%, respectively) against TMV were more potent than that of ningnanmycin (56.2%).

To confirm the potential inhibitory capacity of these compounds against TMV, on the basis of our previous bioassays, we further evaluated the EC_50_ of several target compounds against TMV. As it is evident from Table 4, the antiviral curative activities of compounds **3a**, **3c**, **3l**, and **3s**, with corresponding EC_50_ values of 270.0, 335.4, 238.8, and 250.0 μg/mL against TMV, were much better than that of ningnanmycin (386.2 μg/mL). The protective activities of **3h** and **3j** (EC_50_ of 104.2 and 290.0 μg/mL) against TMV were better than or near to that of ningnanmycin (297.1 μg/mL). 

### 2.5. Structure–Activitiy Relationship (SAR) of the Title Compounds against TMV

As indicated in Table 3 and Table 4, most of phosphorylated penta-1,4-dien-3-one derivatives showed significant antiviral activities against TMV, and some structure–activity relationships can be analyzed and summarized. The presence of 4-Cl-Ph, 4-OCH_3_-Ph, -Ph, and 3-CH_3_-Ph groups at the R_1_ position greatly improved the curative activities of the title compounds against TMV. For example, the title compounds **3a** (4-Cl-Ph), **3c** (4-OCH_3_-Ph), **3l** (-Ph), and **3s** (3-CH_3_-Ph) exhibited significant antiviral activities against TMV, with inhibition rates of 61.2, 62.2, 69.3, and 63.4%, respectively. These compounds were found to be more active compared to other tested compounds. In addition, when R_1_ was 3-NO_2_-Ph, 4-NO_2_-Ph, and 3-CF_3_-Ph groups, the protective activities of the corresponding compounds **3h**, **3j**, and **3p** at 500 μg/mL were 66.7, 63.8, and 58.8%, respectively, which were better than those of other substituent groups. 

### 2.6. Binding Studies of TMV Coat Protein (CP) and ***3h*** or ***3j***

Due to the excellent protective activities of compounds **3h** and **3j**, we speculated that they may interact well with TMV CP. Hence, we studied the interactions between **3h** (or **3j**) and TMV CP using microscale thermophoresis (MST) [35,36]. The affinity curves are shown in Figure 3; compounds **3h** and **3j** showed strong binding with TMV CP. MST measurements showed that **3h** and **3j** bind to TMV CP with dissociation constants (K_d_) of 0.028 ± 0.014 and 0.23 ± 0.09 μmol/L, which were far greater than that of ningnanmycin (K_d_ = 0.52 ± 0.25). Based on anti-TMV activities and MST results, we speculate that the TMV CP protein was a potential target of **3h** (or **3j**).

### 2.7. Molecular Docking of ***3h*** or ***3j*** and TMV CP

To identify the **3h** and **3j** recognition sites in TMV CP (Protein Data Bank (PDB) code: 1EI7), we performed molecular docking using the Gold method with 200 cycles [35,36,37]. As depicted in Figure 4, the two compounds were well embedded in the activity pocket (ARG-46, GLN-34, ARG-90, etc.) between the two subunits of TMV CP. Among them, ARG-46 had a strong hydrogen bond with **3h** (2.022 Å), and NO_2_ (**3h**) demonstrated three hydrogen bonds with GLN-34 (O–H = 1.990 Å, 2.042 Å, and 2.045 Å). Moreover, ARG-46 showed two hydrogen bonds with **3j** (1.848 Å, 2.063 Å), and phosphate demonstrated a strong hydrogen bond with ARG-90 (O–H = 1.927 Å). There were also two hydrogen bonds between the NO_2_ and the residue ARG-90 (O–H = 2.557 Å, 1.942 Å). From the structural analysis, we can find that, when the benzene ring contained NO_2_, the corresponding compounds had better antiviral activity (protective), potentially due to the O on the NO_2_ forming hydrogen bonds (O–H) with GLN-34 and ARG-90, and potentially enhancing the interaction between the small molecules and TMV CP. These analyses indicated that novel phosphorylated penta-1,4-dien-3-one derivatives may be a potential lead structure for developing novel anti-TMV agents.

## 3. Experimental

*Instruments.* All solvents and reagents were purchased from Shanghai Titan Scientific Co., Ltd. (Shanghai, China), were of analytical reagent grade or chemically pure, and were treated with standard methods prior to use. The reactions were monitored by thin-layer chromatography on silica gel GF_254_. Melting points (m.p.) of all synthesized compounds were determined when left untouched on an XT-4-MP apparatus from Beijing Tech. Instrument Co. (Beijing, China). Using tetramethylsilane (TMS) as the internal standard and chloroform as the solvent, ^1^H, ^13^C, and ^31^P nuclear magnetic resonance (NMR) spectra were recorded on a Bruker Ascend-400 spectrometer (Bruker, Germany) and JEOL-ECX 500 NMR spectrometer (JEOL, Tokyo, Japan) operated at room temperature. High-resolution mass spectral (HRMS) data were performed with Thermo Scientific Q Exactive (Thermo, Waltham, MA, USA). The micro thermophoresis of the compound and TMV CP was determined using a micro thermophoresis instrument (NanoTemper Tchnologies GmbH, München, Germany); the fluorescence spectroscopy of the compound interacting with TMV CP was determined using a FluoroMax-4 fluorescence spectrometer (HORIBA Scientific, Palaiseau, France).

### 3.1. Chemistry

#### 3.1.1. Synthesis Procedure for Intermediate **1**


Aqueous sodium hydroxide solution (10% NaOH, 9.0 mmol) was added to a round-bottomed flask containing 2-hydroxybenzaldehyde (8.2 mmol) and acetone (6 mL). The mixture was stirred at ambient temperature for 12 h. The resulting dark-yellow mixture was acidified by HCl (10% HCl, 9.0 mmol) after the reaction was completed. Finally, the mixture was filtered under vacuum, and the residue was dried to yield (*E*)-4-(2-hydroxyphenyl)but-3-en-2-one **1** (1.0 g) [15,29]. 

#### 3.1.2. General Procedure for Preparation of Intermediate **2**


To a round-bottomed flask containing (*E*)-4-(2-hydroxyphenyl)but-3-en-2-one (1.6 g, 0.01 mol) and aromatic aldehyde (0.011 mol) in ethanol (15 mL), a solution of NaOH (10% NaOH, 12 mL) was added. The reaction mixture was stirred for 10 h at ambient temperature. Then, the mixture was diluted with water (200 mL) and neutralized with aqueous HCl (10% HCl, 12 mL). The resulting precipitate was separated and recrystallized from ethanol to yield the penta-1,4-diene-3-ones **2** as yellow solids.

#### 3.1.3. General Procedure for Preparation of Title Compounds **3a**−**3s**


A solution of intermediate **2** (1.0 mmol) and Et_3_N (3.0 mmol) in CCl_4_ (30 mL) was stirred until dissolved. The mixture was stirred in an ice-water bath for 30 min, after adding the mixture of phosphonate (4.0 mmol) and CCl_4_ (6 mL) dropwise. The mixture was then removed from the ice-water bath and stirring continued at room temperature for 24 h. After the reaction was completed (as indicated by TLC), the solvent was removed under depressurization, and the residue was diluted with EtOAc (3 × 35 mL), and it was washed with 5% HCl (3 × 30 mL) and 5% NaOH (3 × 30 mL), respectively. The organic layer was dried by anhydrous Na_2_SO_4_, the solvent was removed under depressurization, and the residue was purified by column chromatography on silica gel with a mixture (V(petroleum ether):V(ethyl acetate) = 2:1) to yield the title compounds **3a**–**3s**. Representative data for **3a** are listed below.

2-((1*E*,4*E*)-5-(4-chlorophenyl)-3-oxopenta-1,4-dien-1-yl)phenyl dimethyl phosphate (**3a**): yellow solid; m.p. 114–115 °C, yield 48%. ^1^H-NMR (500 MHz, CDCl_3_) δ 8.00 (d, *J* = 16.1 Hz, 1H, Ar(2-O)–CH=), 7.67 (s, 1H, Ar-H), 7.64 (d, *J* = 9.8 Hz, 1H, Ar-H), 7.51 (s, 1H, Ar-H), 7.49 (s, 1H, Ar-H), 7.38–7.35 (m, 2H, Ar-H, 4-Cl)-CH=), 7.35 (d, *J* = 2.0 Hz, 1H, Ar-H), 7.33 (d, *J* = 1.9 Hz, 1H, Ar-H), 7.22–7.17 (m, 1H, Ar-H), 7.08 (d, *J* = 16.0 Hz, 1H, Ar(4-Cl)–C=CH), 7.00 (d, *J* = 16.0 Hz, 1H, Ar(4-Cl)–C=CH), 3.87 (s, 3H, CH_3_), 3.85 (s, 3H, CH_3_).^13^C-NMR (126 MHz, CDCl_3_) δ 188.71, 149.63, 149.58, 142.22, 136.95, 136.53, 133.28, 131.86, 129.65, 129.34, 128.14, 127.07, 126.59, 126.53, 125.93, 125.63, 120.77, 55.32, 55.28. ^31^P-NMR (202 MHz, CDCl_3_) δ −3.75. HRMS calculated for C_19_H_19_ClO_5_P [M + H]^+^ 393.0653, found 393.0645.

### 3.2. Bioassays: Antibacterial Bioassays

The antibacterial effects of the target compounds against Xac (strain 29-1, Shanghai Jiao Tong University, Shanghai, China) and Xoo (strain PXO99A, Nangjing Agricultural University, Nangjing, China) were evaluated by the turbidimeter test [30,31,32]. The title compounds were dissolved in 150 μL of dimethyl sulfoxide (DMSO) and diluted with water containing Tween-20 (0.1%) to obtain final concentrations of 100 and 50 μg/mL. DMSO in sterile distilled water served as a blank control, and the commercial agent thiadiazole-copper was used as positive control. Approximately 1 mL of the sample liquid was added to a 15-mL tube, with 4 mL of nutrient broth (NB, 1.5 g of beef extract, 0.5 g of yeast powder, 5.0 g of glucose, 2.5 g of peptone, and 500 mL of distilled water, pH 7.0 to 7.2) media. Then, approximately 40 μL of Xoo or Xac bacterium solution was added. The inoculated test tubes were incubated at 28 ± 1 °C and continuously shaken at 180 rpm for 24–48 h. The growth of the cultures was monitored with a spectrophotometer by measuring the optical density at 595 nm (OD_595_) given by corrected turbidity values. The relative inhibitory (*I*%) values were calculated using the formula below, where *C*_tur_ represents the corrected turbidity value (OD_595_) of bacterial growth on untreated NB (blank control), and *T*_tur_ represents the corrected turbidity values (OD_595_) of bacterial growth on treated NB. 

Inhibition rate *I* (%) = (*C*_tur_ − *T*_tur_)/*C*_tur_ × 100.(1)

Several of the target compounds were tested against Xoo and Xac at five double-declining concentrations (100, 50, 25, 12.5, and 6.25 μg/mL), and their corresponding EC_50_ values were obtained via software SPSS 17.0 (SPSS, Chicago, IL, USA). Each experiment was computed at least three times.

### 3.3. Bioassays: Antiviral Bioassays

#### 3.3.1. Purification of TMV

The upper leaves of *N. tabacum* cv. K326 were selected and inoculated with TMV, using previously reported methods for TMV purification [38].

#### 3.3.2. Curative Activities of Compounds against TMV In Vivo

Using *N. tabacum* L. leaves of the same age as the test subjects, the TMV virus was dipped and inoculated on the whole leaves, which were scattered with silicon carbide beforehand [33,34]. After about 60 min, the leaves were washed with water, and, after drying, the compound solution was smeared onto the left side of leaf, and the solvent was smeared onto the right side. The local lesion numbers were counted after three to four days. Each compound was tested three times.

#### 3.3.3. Protective Activities of Compounds against TMV In Vivo

The compound solution was smeared on the left side of leaf, while the solvent was smeared on the right side [33,34]. The leaves were inoculated with virus 12 h later. Then, the leaves were washed with water after inoculation for 2 h. The local lesion numbers were counted after three to four days. Each compound was conducted three times.

### 3.4. Interaction Studies between ***3h*** or ***3j*** and TMV CP

According to a previous classic method, the binding was calculated for MST Monolith NT.115 (Nano Temper Technologies, Germany) [35,36]. Using NT-647 dye (Nano Temper Technologies, Germany) 0.5 µM purified recombinant proteins were incubated with a range of ligands from 0 µM to 5 µM for 5 min, using the final concentration of 20 nM in the thermophoresis experiment. The selected compounds in DMSO were made into a 16-point dilution series. Each compound dilution series was subsequently transferred to protein solutions (10 mM Tris-HCl and 100 mM sodium chloride pH 7.5, 0.05% Tween-20). The labeled TMV CP with each dilution point (1:1 mix) was incubated for 15 min at ambient temperature; then, the standard capillaries (NanoTemper Technologies, Germany) were added to the samples. Under a setting of 20% light-emitting diode (LED) and 40% infrared (IR) laser, the Monolith NT.115 microscale thermophoresis system (NanoTemper Technologies, Germany) was used to measure interactions. Laser on-time was set at 30 s, and laser-off time was set at 5 s. Using the mass action equation in the Nano Temper software (NT. 115), the K_d_ values were calculated from the duplicate reads, with each experiment being the average of three replicates.

### 3.5. Molecular Docking Study

Molecule docking studies were obtained using DS-CDoking implemented in Discovery Studio (version 4.5, San Diego, CA, USA). Through the UniProt database, we searched for the coat protein subunit amino-acid sequence of TMV. The protein basic local alignment search tool (BLAST) server was used to search the template protein and homologies of the TMV CP sequence so as to align them. Homology modeling of TMV CP was constructed using Create Homology Models, which is a model integrated in Discovery Studio, using Ramachandran plots to evaluate the obtained models. The three-dimensional (3D) structures of the compounds were carried out using the Sketching module and were optimized using the Full Minimization module. During the docking process, all parameters were defaults [35,36,37]

## 4. Conclusions 

In summary, with the aim of developing a novel, highly efficient, and environmentally benign virucide, we introduced a penta-1,4-diene-3-one group into phosphate derivatives to synthesize 19 curcumin derivatives. Their antibacterial activities against Xac and Xoo in vitro and their antiviral activity against TMV in vivo were evaluated. Bioassay results showed that several of the title compounds exhibited good antibacterial and antiviral activities. In particular, compounds 3h and 3j showed remarkable protective activity against TMV. Furthermore, the microscale thermophoresis showed that compounds 3h and 3j have strong binding capability with TMV CP, and the molecular docking studies were consistent with the experimental results. Given the above results, these novel phosphorylated penta-1,4-dien-3-one derivatives should be further studied as potential alternative templates in the search for novel antibacterial and antiviral agents.

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
