# Peer review of "Novel Phosphorylated Penta-1,4-dien-3-one Derivatives: Design, Synthesis, and Biological Activity"

_molecules, 2019, doi:10.3390/molecules24050925_

Round 1

Reviewer 1 Report

The authors should at least consider the following suggestions/corrections:

1. Keywords: penta-1,4-dien-3-ones; phosphorylation; .......

2. In the Introduction section of the paper:

l. 30: Bacterial diseases, such as ........, caused by .......

l. 40-41: The figure should be improved (better analysis) because it is in some places blurred

l. 51: Phosphonates and their immediate derivatives...

        Note: please explain what do you mean by immediate derivatives)

l. 52 and 53: Figure 1.b, Figure 1.c

l. 54: ...., they possess improved (or higher) cell permeability, ....

l. 59-60: .....for developing highly (efficient????) agrochemicals.... (Note: something is missing here)

l. 60-61: ....a penta-1,4-dien-3-one group into phosphate (not phosphonate) derivatives,..... novel phosphate (not phosphonate) ....... Thus, 18 (Ι count 19!!) novel.......

3. In the Results and Discussion (Chemistry) Section:

l. 66: ...............schematized depicted.............

l. 68: ....at normal temperature to obtain....  ....at ambient temperature to yield (or provide)....

l. 71-72: ...by condensing intermediates 2 and dialkyl phosphonate (or dialkyl phosphite) in the presence of Et3N in CCl4 at ambient temperature for 24 h.

l. 76: ...........nucleuses nuclei, and a singlet at δ 3.87-3.85 ppm indicates............... and two singlets at δ 3.87 and 3.85 ppm indicate .........

l. 80: 2.2 Antibacterial activity of title compounds ...........

l. 84-50: The analysis of the figure (Scheme 1) should be improved

l. 91: activities........67.9%, respectively, which was

l. 93:  Table 1...................................compounds (3a-3s) ..........

l. 98: in Table 2. Compounds.....

l. 121: Table 2.  of the selected title compounds.......

            Xoo is missing from the first line of the table and Xac should be centralized over the 5-7 columns

l. 133-134: At the bottom of the table Ningnanmycinb

                  At the footnote of the table:  aAverage of three replicates. bThe commercial antiviral agent Ningnanmycin.

l. 136-137: .....against TMV. In As it is evident from Table 4, ........

l. 152-153: curative  Curative

                                  EC50a

                                   Ningnamycy Ningnamycinb

                 At the footnote of the table:  aAverage of three replicates. bThe commercial antiviral agent Ningnanmycin.

lines 154, 162, 195, 201, 207 and 259 (also throughout the text): the numbers for intermediates 1 and 2 and compounds 3a to 3s should be in bold face

l. 156: Please explain what TMV-CP stands for. Throughout the text, please either use TMV-CP or TMV CP but not both.

l. 158: ...... bond to....  .......bind to ....

l. 160-161: Based on........, we speculate that.......

l. 164: ...... Figure 4.......

l. 171-173: needs improvement-not easily understood what the authors imply here

l. 170 and 172: .... we can found find..... it may enhanced enhance.......

l. 196: What is the normality of the NaOH solution used?

l. 198-200: needs improvement

l. 203: ......a solution (??? mL) of NaOH was added.....

l. 206: .......to yield the penta-1,4-diene-3-ones 2 as yellow solids.

l. 215: .......and the residue was purified by column chromatography on silica gel to obtain the title compounds 3a-3s.

 Note: Neither here nor in the supporting information section are provided the eluant(s) for such purifications and the Rf values for the purified compounds. Please provide this information.

l. 229: ..... dimethylformamide (DMSO)... which of the two is correct? Please correct accordingly.

l. 274-275: Before docking, the protein were was treated by removing water and  hydrogenation (?? please explain). Molecular docking were was then......

l. 275-277: needs improvement

4. In the Conclusions section of the paper:

l. 280: ..... into phosphonate derivatives..... into phosphate derivatives... [please correct where necessary in the text as the synthesized esters are of the phosphate (actually, they are mixed phosphates) and not of the phosphonate type]

5. In the References section of the paper:

l. 308: .......Rice Diseases........

l. 312: ....Citrus canker-A review. J. Appl. Hortic. 1997,......

l. 315: ....... Ann. Appl. Biol.......

General note: All journals should be presented as above (l. 312 and 315), as can be seen from any recent published paper in Molecules. Further examples: Chin. J. Org. Chem.    ......     Bioorg. Med. Chem. Lett.  ........etc

l. 310: ...Khust, G.S.,

l. 327: .......... 4(3H)-.....

l. 389: ...... 2015,....

l. 394: .......2018,....

l. 395:   ......Samples of the compounds ......  (which ones??. If all, please indicate it as e.g. ...Samples of all synthesized compounds) are available from the authors.

A general note: The authors seem to have uploaded two additional but identical pdf files, one titled non-published (1) and the other one titled Supplementary (1). What is the point of it?

Author Response

Thank you very much for your letter and the comments from the referees about our paper entitled “Novel phosphorylated penta-1,4-dien-3-one derivatives: Design, synthesis, and biological activity ” (ID molecules-452942). The manuscript has been modified by carefully following the editor and reviewers comments. Each comment has been taken care of and answered by referring to the relevant part of the manuscript. All of the reviewers’ comments have been addressed. These revisions marked with yellow background in the manuscript and also listed below for your reference. Attached please find the revised version, which we would like to submit for your kind consideration. We would like to express our great appreciation to you and reviewers for comments on our paper. Looking forward to hearing from you.

Yours sincerely,

Wei Xue

Below is the address of the corresponding author:

Prof. Wei Xue

Guizhou University, Huaxi District, Guiyang, Guizhou Province

P. R. China 550025

Fax: 0086-0851-88292920  E-mail: [email protected]

Responds to Reviewer 1:

1. Keywords: penta-1,4-dien-3-ones; phosphorylation;

Answer: Thank you for your suggestion. The key word “penta-1,4-dien-3-one; phosphorylated” was changed as “penta-1,4-dien-3-ones; phosphorylation” on my manuscript. Please refer to the modified details in the revised version.

2. In the Introduction section of the paper:

        1. 30:Bacterial diseases, such as......,caused by......

Answer: Thank you for your suggestion. The sentence “Pathogenic bacteria, such as rice bacterial leaf blight and citrus canker caused by the pathogens Xanthomonas oryzae pv. oryzae (Xoo)” was changed as “Bacterial diseases, such as rice bacterial leaf blight and citrus canker, caused by the pathogens Xanthomonas oryzae pv. oryzae (Xoo)”.

1. 40-41: The figure shoud be improved (better analysis) because it as some blurred.

       Answer: Thank you for your suggestion. The figure has been improved.

      1. 51:Phosphonates and their immediate derivatives .....Note:please explain        what do you mean by immediate derivatives).

Answer: Thank you for your suggestion. The sentence “Phosphonates and its immediate derivatives” was changed as “Phosphonates and their immediate derivatives”, and the mean of the “immediate derivatives” was “the analogs derivatives which containing the phosphate groups”. Please refer to the modified details in the revised version.

1. 52 and 53: Figure 1.b, Figure 1.c

Answer: Thank you for your suggestion. The “Figure1.c” was changed as

“Figure 1.c” . Please refer to the modified details in the revised version.

1. 54:.....they possess improved (or higher) cell permeability, ......

Answer: Thank you for your suggestion. The sentence “it possesses more cell permeability” was changed as “they possess improved cell permeability”. Please refer to the modified details in the revised version.

1. 59-60:......for developing highly (efficient????) agrochemicals.....(Note:something is missing here)

Answer: Thank you for your suggestion. The word “efficient” has been added. Please refer to the modified details in the revised version.

1. 60-61:....a penta-1,4-dien-3-one group into phosphate (not phosphonate) derivatives ........novel phosphate (not phosphonate)....Thus, 18 ( I count 19 !!!) novel.....

Answer: Thank you for your suggestion. The sentence “a penta-1,4-diene-3-one groups into phosphonate derivatives, which might generate novel phosphonate derivatives with potent biological activities. Thus, 18 novel phosphorylated” was changed as “a penta-1,4-diene-3-one group into phospate derivatives, which might generate novel phosphate derivatives with potent biological activities. Thus, 19 novel phosphorylated”. Please refer to the modified details in the revised version.

 3.In the Results and Discussion (Chemistry) Section:

            l. 66: ...............schematized depicted.............

Answer: Thank you for your suggestion. The word “schematized” was changed as “depicted”. Please refer to the modified details in the revised version.

             l. 68: ....at normal temperature to obtain........at ambient temperature to                    yield (or provide)....

Answer: Thank you for your suggestion. The sentence “at normal temperature to obtain” was changed as “at ambient temperature to yield”. Please refer to the modified details in the revised version.

             l. 71-72: ...by condensing intermediates 2 and dialkyl phosphonate (or                      dialkyl phosphite) in the presence of Et3N in CCl4 at ambient temperature              for 24 h.

Answer: Thank you for your suggestion. The sentence “via substitution of intermediates 2 and phosphonate with Et3N in CCl4 at normal temperature for 24 h” was changed as “.by condensing intermediates 2 and dialkyl phosphonate (or dialkyl phosphite) in the presence of Et3N in CCl4 at ambient temperature for 24 h.” Please refer to the modified details in the revised version.

1. 76……nucleuses nuclei, and a singlet at δ 3.87-3.85 ppm indicate……and two singlets at δ 3.87 and 3.85 ppm indicate…..

Answer: Thank you for your suggestion. The word “nucleuses” was changed as “nuclei”. The sentence “a singlet at δ 3.87-3.85 ppm indicates” was changed as “two singlets at δ 3.87 and 3.85 ppm indicate”. Please refer to the modified details in the revised version.

            l. 80: 2.2 Antibacterial activity of title compounds ...........

Answer: Thank you for your suggestion. The sentence “Antibacterial Activity of Title Compounds” was changed as “Antibacterial activity of title compounds”. Please refer to the modified details in the revised version.

            l. 84-50: The analysis of the figure (Scheme 1) should be improved

Answer: Thank you for your suggestion. The figure (Scheme 1) has been improved. Please refer to the modified details in the revised version.

            l. 91: activities........67.9%, respectively, which was

Answer: Thank you for your suggestion. The sentence “The antibacterial activitie of compound 3e against Xac at 100 and 50 μg/mL were 77.6 and 67.9%, which was superior to those of TC (57.2 and 27.8%) and BT (70.3 and 54.9%). was changed as “The antibacterial activities of compound 3e against Xac at 100 and 50 μg/mL were 77.6 and 67.9%, respectively, which was superior to those of TC (57.2 and 27.8%) and BT (70.3 and 54.9%)”. Please refer to the modified details in the revised version.

             l. 93: Table 1...................................compounds (3a-3s) ..........

Answer: Thank you for your suggestion. The word “3r was changed as “3s”. Please refer to the modified details in the revised version.

            l. 98: in Table 2. Compounds.....

Answer: Thank you for your suggestion. The word “Tables was changed as “table”. Please refer to the modified details in the revised version.

              l. 121: Table 2. Of the selected title compounds.......

            Xoo is missing from the first line of the table and Xac shoud be centralized               over the 5-7 columns. 

Answer: Thank you for your suggestion. The word “the” was changed as “selected”. The word “Xoo” was added to the manuscript and Xac has been centralized over the 5-7 columns. Please refer to the modified details in the revised version.

1.136-137:.....against TMV. In As it is evident from Table 4.....

Answer: Thank you for your suggestion. The sentence “In Table 4” was changed as “As it is evident from Table 4”. Please refer to the modified details in the revised version.

1.133-134: At the bottom of the table Ningnanmycinb

At the footnote of the table:aAverage of three replicates. bThe commercial antiviral agent Ningnanmycin.

Answer: Thank you for your suggestion. The sentence “Average of three replicates. aThe commercial antiviral agent Ningnanmycin” was changed as “aAverage of three replicates. bThe commercial antiviral agent Ningnanmycin”. Please refer to the modified details in the revised version.

At the At the footnote of the table: aAverage of three replicates. bThe commercial antiviral agent Ningnanmycin.

Lines 154,162, 195, 201, 207 and 259 (also throughout the text): the numbers for intermediates 1 and 2 and compounds 3a to 3s shoud be in bold face.

Answer: Thank you for your suggestion. The word “curative” was changed as “Curative”. “Ningnamycin” was changed as “Ningnamycinb”. The footnote was changed as “aAverage of three replicates. bThe commercial antiviral agent Ningnanmycin”. The numbers for intermediates 1 and 2 and compounds 3a to 3s (throughout the text) has been in bold face. Please refer to the modified details in the revised version.

1. 156: Please explain what TMV-CP stands for. Throughout the text, please either use TMV-CP or TMV CP but not both.

Answer: Thank you for your suggestion. TMV CP was the tobacco mosaic virus coat protein. Throughout the text, “TMV-CP” was changed as “TMV CP”. Please refer to the modified details in the revised version.

             l. 158: ......bond to....  .......bind to ....

              Answer: Thank you for your suggestion. The word “bond” was changed as               "bind".

l. 160-161: Based on........, we speculate that........

Answer: Thank you for your suggestion. The word “that” has been added on my manuscript. Please refer to the modified details in the revised version.

1. 164:.....Figure 4.....

Answer: Thank you for your suggestion. The word “figure 4” was changed as

“Figure 4”. Please refer to the modified details in the revised version.

           l. 171-173: needs improvement-not easily understood what the authors                     imply here

Answer: Thank you for your suggestion. I have used a professional language editing service to ensure the quality of my work. The rewritten section are as followed: From the structural analysis we can find that when the benzene ring existence “NO2”, the corresponding compounds have better antiviral activity (protective), it may be due to the “O” on the “NO2” forms hydrogen bond (O-H) with GLN-34 and ARG-90, and it may enhance the interaction between the small molecules and the TMV-CP. Please refer to the modified details in the revised version.

            l. 170 and 172: .... we can found find..... it may enhanced enhance.......

Answer: Thank you for your suggestion. The sentence “we can found that when the benzyl existence “NO2” is the best for TMV may be due to the “O” on the “NO2” forms O-H hydrogen bond with GLN-34 and ARG-90, and it may enhanced the interaction between the small molecules and the TMV-CP.” was changed as “we can find that when the benzene ring existence “NO2”, the corresponding compounds have better antiviral activity (protective), it may be due to the “O” on the “NO2” forms hydrogen bond (O-H) with GLN-34 and ARG-90, and it may enhance the interaction between the small molecules and the TMV-CP.” Please refer to the modified details in the revised version.

            l. 196: What is the normality of the NaOH solution used?

Answer: Thank you for your suggestion. The normality of the NaOH solution is 10 % (2.78 mol/L), and I have added on my manuscript. Please refer to the modified details in the revised version.

            l. 198-200: needs improvement

Answer: Thank you for your suggestion. I have used a professional language    editing service to ensure the quality of my work. The rewritten section are as followed: Aqueous sodium hydroxide solution (10% NaOH, 9.0 mmol) was added to a round-bottomed flask containing 2-hydroxybenzaldehyde (8.2 mmol) and acetone (6 mL). The mixture was stirred at ambient temperature for 12 h. The resulting dark yellow mixture was acidified by HCl (10% HCl, 9.0 mmol) after the reaction was completed. Finally, the mixture was filtration under vacuum and the residue dried to yield (E)-4-(2-hydroxyphenyl)but-3-en-2-one 1 (1.0g). [13, 27]. Please refer to the modified details in the revised version.

            l. 203: ......a solution (??? mL) of NaOH was added.....

Answer: Thank you for your suggestion. The dosege of the NaOH solution is 10% 12 mL, and I have added on my manuscript. Please refer to the modified details in the revised version.

             l. 206: .......to yield the penta-1,4-diene-3-ones 2 as yellow solids.

Answer: Thank you for your suggestion. The sentence “to obtain the yellow solid penta-1,4-diene-3-one 2” was changed as “to yield the penta-1,4-diene-3-ones 2 as yellow solids”. Please refer to the modified details in the revised version.

1. 215 …. and the residue was purified by column chromatography on silica gel to obtain the title compounds 3a-3s.

Note: Neither here nor in the supporting information section are provided the eluant(s) for such purifications and the Rf values for the purified compounds. Please provide this information.

Answer: Thank you for your suggestion. The sentence “and the residue was purified by column chromatography on silica gel to obtain the title compounds 3a-3s.” was changed as “and the residue was purified by column chromatography on silica gel with a mixture [V (petroleum ether): V(ethyl acetate)=1:1] to yield the title compounds 3a-3s.” The Rf values for the purified compounds was added in the supporting information. Please refer to the modified details in the revised version.

1. 229:.....dimethylformamide (DMSO)...which of the two is correct? Please correct accordingly.

Answer: Thank you for your suggestion. The word “dimethylformamide” was changed as “dimethyl sulfoxide”. Please refer to the modified details in the revised version.

            l. 274-275: Before docking, the protein were was treated by removing water            and hydrogenation (?? please explain). Molecular docking were was                        then......

Answer: Thank you for your suggestion. The word “were” was changed as “was” , before use, the protein is stored in deionized water, so before docking, the protein was treated by remove water and hydrogenation was to prevent deviation.

            l. 275-277: needs improvement

Answer: Thank you for your suggestion. I have used a professional language editing service to ensure the quality of my work. The rewritten section are as followed: Molecule docking study were obtained by using DS-CDoking implemented in Discovery Studio (version 4.5). Through the UniProt database, we searching the coat protein subunit amino acid sequence of TMV. Using the protein BLAST server to search the template protein and homologies of TMV CP sequence were aligned. Homology modeling of TMV CP was constructed using Create Homology Models, which is a model integrated in Discovery Studio. Using Ramachandran plots evaluate the obtained models. The 3D structures of the compounds were carried out using the Sketching module and optimized by the Full Minimization module. During the docking process all parameters were default. Please refer to the modified details in the revised version.

4. In the Conclusion section of the paper:

             1. 280:….into phosphonate derivatives….into phosphate derivatives….[please                  correct where necessary in the text as the synthesized esters are of the phosphate                    (actually, they are mixed phosphates) and not of the phosphonate type].

Answer: Thank you for your suggestion. The word “phosphonate” was changed as “phosphate”. Please refer to the modified details in the revised version.

5. In the Refenrences section of the paper:

1.308:….Rice Diseases….

1.312…..Crirus canker-A review. J. Appl. Hortic. 1997,…..

1.315:……Ann. Appl. Biol…..

General note: All journals shoud be presented as above (1.312 and 315), as can be seen from any recent published paper in Molecules. Further examples: Chin. J. Org. Chem.    …Bioorg. Med. Chem. Lett….etc.

1.310……Khust, G.S.

1.327:……4(3H)-…..

1.389…..2015….

1.394…..2018……

Answer: Thank you for your suggestion. The references I was revised according to the demand. Please refer to the modified details in the revised version.

            l. 395:.....Samples of the compounds ......(which ones??.If all, please                        indicate it as e.g. ..Samples of all synthesized compounds) are available                  from the authors.

             Answer: Thank you for your suggestion. Samples of all synthesized compounds  3a-3s  are avilable from the authors.

In addition, we tried our best to improve the manuscript and made some changes in the manuscript. These changes will not influence the content and framework of the paper. And here we did not list the changes but marked with yellow background in revised paper.

We appreciate for Editors/Reviewers’ warm work earnestly, and hope that the correction will meet with approval.

Reviewer 2 Report

1. The introduction is rather bland and very stereotypical, I recommend re-writing it.  

1. The text-caption under the structures in figure 2 is not easily readable.

2. The R substituents in scheme 1 are also hard to read, I suggest changing font size or style.

3. Specify what are TC and BT at line 83; EC in line 97;

4. In the 2.5. Molecular docking..... paragraph please note that all predicted binding mode interactions are hypothetical, and should be described as such. As the text reads now, one has the impression that these interactions were determined via X-ray crystallography of the active substance-target complex.   

5. The Conclusion paragraph should be completely re-written. In the present form a lot of results from the biological activity assessment is provided and no mention of the SAR observation or docking results. Also the last paragraph is too generalizing and provides no actual information.

6. Considering the References I find it troublesome that some are rather old. Please update them and try to cite relevant work from this millennium.   

5. Please consider rephrasing the following:

Line 32 “strongly restrain the agricultural output worldwide and are difficult to control in agriculture.”

Line 51 „its”

Line 60 “highly agrochemicals”

Line 67 „is schematized in 67 Scheme 1”

Line 72 „normal temperature”

Line 81 „bacterial”

Line 91 activitie

Line 115 bioactivitie

Line 128 from weak to good

Line 170 we can found

Line 171 when the benzyl existence “NO2” is the best for TMV

Line 199 Finally filtration and dried

Line 204 a solution of NaOH was added in the form of 10% aqueous solution

Line 280 groups

Concerning English use, overall the paper has a rough-feel to it so I recommend having it re-written by a native speaker.  

Author Response

Thank you very much for your letter and the comments from the referees about our paper entitled “Novel phosphorylated penta-1,4-dien-3-one derivatives: Design, synthesis, and biological activity ” (ID molecules-452942). The manuscript has been modified by carefully following the editor and reviewers comments. Each comment has been taken care of and answered by referring to the relevant part of the manuscript. All of the reviewers’ comments have been addressed. These revisions marked with yellow background in the manuscript and also listed below for your reference. Attached please find the revised version, which we would like to submit for your kind consideration. We would like to express our great appreciation to you and reviewers for comments on our paper. Looking forward to hearing from you.

Yours sincerely,

Wei Xue

Below is the address of the corresponding author:

Prof. Wei Xue

Guizhou University, Huaxi District, Guiyang, Guizhou Province

P. R. China 550025

Fax: 0086-0851-88292920  E-mail: [email protected]

Responds to Reviewer 2:

     1. The introduction is rather bland and very stereotypical, I recommend 

         re-writing it.

Answer: Thank you for your suggestion. I have used a professional language editing service to ensure the quality of my work. The rewritten introduction are as followed:   Please refer to the modified details in the revised version.

     2. The text-caption under the structures in figure 2 is not easily readable.

Answer: Thank you for your suggestion. The text-caption under the structures in figure 2 I has been improved. Please refer to the modified details in the revised version.

3.  The R substituents in scheme 1 are also hard to read, I suggest changing font size or style.

Answer: Thank you for your suggestion. I have changed the font size and style of scheme 1. Please refer to the modified details in the revised version.

     4.  Specify what are TC and BT at line 83; EC in line 97;

Answer: Thank you for your suggestion. TC, BT and EC50 were described briefly in abstract section, thiodiazole-copper (TC), bismerthiazol (BT), 50% effective concentration (EC50). Please refer to the modified details in the revised version.

5In the 2.5. Molecular docking..... paragraph please note that all predicted binding mode interactions are hypothetical, and should be described as such. As the text reads now, one has the impression that these interactions were determined via X-ray crystallography of the active substance-target complex.

Answer: Thank you for your suggestion. The molecular docking section I have revised. The rewritten introduction are as followed: Molecule docking study were obtained by using DS-CDoking implemented in Discovery Studio (version 4.5). Through the UniProt database, we searching the coat protein subunit amino acid sequence of TMV. Using the protein BLAST server to search the template protein and homologies of TMV CP sequence were aligned. Homology modeling of TMV CP was constructed using Create Homology Models, which is a model integrated in Discovery Studio. Using Ramachandran plots evaluate the obtained models. The 3D structures of the compounds were carried out using the Sketching module and optimized by the Full Minimization module. During the docking process all parameters were default. Please refer to the modified details in the revised version.

6.  The Conclusion paragraph should be completely re-written. In the present form a lot of results from the biological activity assessment is provided and no mention of the SAR observation or docking results. Also the last paragraph is too generalizing and provides no actual information.

Answer: Thank you for your suggestion. I have used a professional language    editing the conclusion paragraph and added the SAR observation or docking results. The rewritten conclusion are as followed: In summary, with the aim to develop a novel, highly-efficient, and environmentally benign virucide, we introduced a penta-1,4-diene-3-one group into phosphate derivatives to synthesize 19 curcumin derivatives. Their antibacterial activities against Xac and Xoo in vitro and their antiviral activity against TMV in vivo were evaluated. Bioassay results showed that several of the title compounds exhibited good antibacterial and antiviral activities. In particular, compounds 3h and 3j showed remarkable protective activity against TMV. Furthermore, the microscale thermophoresis showed that compounds 3h and 3j have strong binding capability with TMV CP, and the molecular docking studies were consistent with the experimental results. Given the above results, these novel phosphorylated penta-1,4-dien-3-one derivatives should be further studied as potential alternative templates in the search for novel antibacterial and antiviral agents. Please refer to the modified details in the revised version.

7.  Considering the References I find it troublesome that some are rather old. Please update them and try to cite relevant work from this millennium.

Answer: Thank you for your suggestion. The references I have revised and update some of them. Please refer to the modified details in the revised version.

8. Please consider rephrasing the following:

Line 32 "strongly restrain the agricultural output worldwide and are difficult to control in agriculture.

Answer: Thank you for your suggestion. The sentence “strongly restrain the agricultural output worldwide and are difficult to control in agriculture” was changed as “are strongly restrain the agricultural output worldwide and are difficult to control in agriculture”. Please refer to the modified details in the revised version.

Line 51 "its"

Answer: Thank you for your suggestion. The word “its” was changed as “their”. Please refer to the modified details in the revised version.

Line 60 "highly agrochemicals.

Answer: Thank you for your suggestion. The sentence “highly agrochemicals” was changed as “highly efficient agrochemicals”. Please refer to the modified details in the revised version.

     Line 67 “is schematized in 67 Scheme 1”

Answer: Thank you for your suggestion. The word “schematized” was changed as “depicted”. Please refer to the modified details in the revised version.

Line 72 "normal temperature"

Answer: Thank you for your suggestion. The word “normal” was changed as “ambient”. Please refer to the modified details in the revised version.

Line 81 "bacterial"

Answer: Thank you for your suggestion. The word “bacterial” was changed as “antibacterial”. Please refer to the modified details in the revised version.

Line 91 activitie

Answer: Thank you for your suggestion. The word “activitie” was changed as “activities”. Please refer to the modified details in the revised version.

Line 115 bioactivitie

Answer: Thank you for your suggestion. The word “biactivitie” was changed as “bioactivities”. Please refer to the modified details in the revised version.

Line 128 from weak to good

Answer: Thank you for your suggestion. The sentence “Compounds 3a to 3r exhibited from weak to good antiviral activities against TMV.” was changed as “All tested compounds exhibited from weak to good antiviral activities against TMV.” Please refer to the modified details in the revised version.

Line 170 we can found

Answer: Thank you for your suggestion. The sentence “we can found” was changed as “we can find”. Please refer to the modified details in the revised version.

Line 171 when the benzyl existence "NO2" is the best for TMV.

Answer: Thank you for your suggestion. The sentence “when the benzyl existence “NO2” is the best for TMV” was changed as “when the benzene ring existence “NO2”, the corresponding compounds have better antiviral activity (protective).” Please refer to the modified details in the revised version.

Line 199 Finally filtration and dried.

Answer: Thank you for your suggestion. The sentence “Finally filtration and dried under vacuum to yield (E)-4-(2-hydroxyphenyl)but-3-en-2-one 1 (1.0g).” was changed as “Finally, the mixture was filtration under vacuum and the residue dried to yield (E)-4-(2-hydroxyphenyl)but-3-en-2-one 1 (1.0g).” Please refer to the modified details in the revised version.

Line 204 a solution of NaOH was added in the form of 105% aqueous solutions.

Answer: Thank you for your suggestion. The sentence “a solution of NaOH was added in the form of 10% aqueous solution” was changed as “a solution of NaOH (10% NaOH, 12 mL) was added”. Please refer to the modified details in the revised version.

Line 280 groups

Answer: Thank you for your suggestion. The word “groups” was changed as “group”. Please refer to the modified details in the revised version.

In addition, we tried our best to improve the manuscript and made some changes in the manuscript. These changes will not influence the content and framework of the paper. And here we did not list the changes but marked with yellow background in revised paper.

We appreciate for Editors/Reviewers’ warm work earnestly, and hope that the correction will meet with approval.

Round 2

Reviewer 2 Report

Thank you for taking into consideration my comments.